# Supra-Versus Submaximal Cycle Ergometer Verification of VO_2max_ in Males and Females

**DOI:** 10.3390/sports8120163

**Published:** 2020-12-12

**Authors:** Brandon J. Sawyer, Nicholas McMahon, Kirsten L. Thornhill, Brett R. Baughman, Jenny M. Mahoney, Kai L. Pattison, Kaitlin A. Freeberg, Ryan T. Botts

**Affiliations:** 1Departments of Kinesiology and Biology, Point Loma Nazarene University, San Diego, CA 92106, USA; jmahoney@pointloma.edu; 2Department of Kinesiology, Point Loma Nazarene University, San Diego, CA 92106, USA; nmcmahon1993@pointloma.edu (N.M.); kthornhi@pointloma.edu (K.L.T.); bbaughman1215@pointloma.edu (B.R.B.); kpattison290@pointloma.edu (K.L.P.); kaitlin.freeberg@colorado.edu (K.A.F.); 3Department of Mathematical, Information, and Computer Sciences, Point Loma Nazarene University, San Diego, CA 92106, USA; rbotts@pointloma.edu

**Keywords:** maximal oxygen uptake, true VO_2max_, criteria for VO_2max_, incremental exercise testing, sex difference

## Abstract

This study was designed to determine the optimal intensity for verification phase testing (VP) in healthy, young adults. Thirty one young, active participants (16 females) completed a cycle ergometer graded exercise test (GXT) VO_2max_ test and 4 VP tests at 80, 90, 100, and 105% of the maximum wattage achieved during the GXT. GXT and VP VO_2max_ values showed a significant test x sex interaction (*p* = 0.02). The males elicited significantly higher VO_2max_ values during the GXT, 80%, and 90% when compared to the 105%, (105 vs. GXT: *p* = 0.05; 105% vs. 80%: *p* < 0.01; 105% vs. 90%: *p* = 0.02). There were no significant differences in VO_2max_ across the tests in the females (*p* > 0.05); 80% of the males achieved their highest VP VO_2max_ during a submaximal VP test compared to only 37.5% of the females. A secondary study conducted showed excellent reliability (ICCs > 0.90) and low variation (CVs < 3%) for the 90% VP. Our findings show that a submaximal verification phase intensity is ideal for young healthy males to elicit the highest VO_2max_ during cycle ergometer testing. For females, a range of intensities (80–105%) produce similar VO_2max_ values. However, the 80% VP yields an unnecessarily high time to exhaustion.

## 1. Introduction

Maximal oxygen uptake (VO_2max_) represents the maximal rate at which oxygen delivery and utilization can occur. VO_2max_ is known as the gold standard measurement of cardiorespiratory fitness and is one of the most common tests in exercise physiology research [1]. VO_2max_ testing has many practical uses including: predicting mortality and disease risk [2], aiding in exercise prescription, tracking changes in cardiorespiratory fitness, testing the efficacy of exercise interventions, as well as many clinical uses in cardiology and pulmonology [3]. The broad use and importance of VO_2max_ testing underscores the necessity for accuracy in determining whether a true VO_2max_ was actually attained. The original primary criterion to ensure VO_2max_ had occurred was attainment of a plateau in oxygen uptake during the final stages of a maximal exercise test [4]. Subsequent studies have shown it to be unnecessary for VO_2_ to plateau to ensure VO_2max_ [5,6]. Most researchers use secondary criteria, including maximal values of blood lactate, heart rate (HR), and/or respiratory exchange ratio (RER) even though these have been shown to have major limitations [7,8,9], high individual variability [10], and may occur well before maximal effort is reached [9].

Due to these issues with the primary and secondary criteria of VO_2max_ attainment, a constant work rate, an exhaustive bout of exercise following the standard graded exercise test (GXT), has been strongly recommended in order to verify that VO_2max_ has been attained [11,12,13]. This verification phase test (VP) shows promise as a solution to the shortcomings of primary and secondary criteria. As has recently been reviewed by Schaun [13], the lack of standardization of VP protocols and criteria for achievement of verified VO_2max_ remain major limitations. The intensity of the VP test relative to the maximum work rate achieved on the GXT is one of the issues that still needs to be resolved.

The most common method of verification phase testing is using a work rate that exceeds that achieved on the GXT (IE: supramaximal work rate) [11,12,13] in order to extend the work rate–VO_2_ relationship and construct a VO_2_ plateau [11]. In contrast, some authors have used a maximal [14] or even submaximal VP work rate [5,15,16,17,18] for verification of VO_2max_ attainment. It is possible that submaximal work rates will lead to longer times to exhaustion and possibly higher VO_2max_ values in some individuals [19], especially those unaccustomed to high-intensity exercise [14]. Some individuals may reach exhaustion in less than 2 min during supramaximal VPs, which may not be enough time to elicit VO_2max_ [19]. A recent study using data from 109 subjects found supramaximal VPs to elicit a significantly lower mean VO_2max_ compared to the GXT [20]. Theoretically, any work rate that falls in the severe domain should lead to exhaustion with VO_2max_ attainment [21,22]. Furthermore, it is possible that some supramaximal work rates used in VP testing may actually exceed the upper limit of the severe domain and enter into the extreme domain where exhaustion occurs before VO_2max_ attainment [19,23]. A few studies have shown submaximal verification tests to elicit VO_2max_ values similar to the GXT [5,15,16,17,18]. While some of these studies have tested both sub- and supramaximal VPs together [16,17,18], only two directly compared VP VO_2max_ values to each other [16,17]. These two studies [16,17] did not find differences in VO_2max_ between the GXT and VP tests, but they had small sample sizes and were not designed to test for a sex difference. Identification of the work rate (whether submaximal, maximal, or supramaximal) that elicits the highest VO_2max_ in both males and females during a cycle ergometer VP test would be helpful in deciding which VP work rate should be used. Furthermore, no study has yet assessed test-retest reliability of a submaximal verification phase test.

Therefore, the primary purpose of this study was to test which VP work rate produces the highest VO_2max_ value. We compared submaximal, maximal, and supramaximal work rates in both males and females in an attempt to determine the optimal work rate for VP testing. Our secondary purpose was to determine whether or not the optimal VP work rate differs between males and females. Finally, as a follow up to our primary study, we also tested the reliability of the 90% VP test.

## 2. Materials and Methods

### 2.1. Subjects

Both the primary study and the reliability study were approved by the Point Loma Nazarene University Institutional Review Board (PLNU IRB ID #17609) and conformed to the ethical standards of the Declaration of Helsinki. All subjects provided written informed consent before participation. All subjects met the following inclusion criteria: free from known chronic disease, between 18 and 30 years of age, completion of the Physical Activity Readiness Questionnaire (PAR-Q) [24] without “yes” answers. For the primary study, on the basis of previously published data [14], using R [25] powerlmm package [26] we calculated that completing 15 subjects would yield 90% power to detect a 4% difference in VO_2max_ between tests (at a two-tailed alpha level of 0.05). For the reliability study we used G power [27] along with previously published data [14] and found that a sample size of 8 was needed to produce 90% power to detect a significant correlation in VO_2max_ between repeated VP tests. For the primary study we enrolled 34 physically active (between 100 and 300 min of moderate or vigorous activity per week) subjects (15 males and 19 females) into this 5-visit randomized cross-over study. For the reliability study we enrolled 23 young healthy physically active (between 100 and 300 min of moderate or vigorous activity per week) males and females to complete a 3-visit study. In each study we enrolled enough subjects to have adequate power within each sex independently.

### 2.2. Experimental Design

Primary Study: All subjects completed the following 5 visits: 1) GXT: standard ramp-style VO_2max_ test; 2–5) randomly assigned verification phase tests at either 80, 90, 100, or 105% of the maximal work rate achieved on the GXT. Each test included a 5 min warm up at 50 W (males) or 30 W (females), followed by the test. All participants were asked to arrive to the lab ~ 4 h having fasted and abstained from caffeine, alcohol, and dietary supplements for at least 24 h and to abstain from vigorous exercise or weight lifting for 48 h before each visit. All visits were conducted at the same time of day for each subject, separated by at least 48 h, and all testing was completed within 3 weeks for each subject.

Reliability Study: Participants completed 3 separate visits to the Exercise Physiology Laboratory. Each visit included a ramp-style GXT test followed by a 10-min active recovery period, then directly into a 90% power output VP test. The first visit was used as a familiarization visit. Therefore, we conducted our reliability analyses on the tests conducted on visits 2 and 3 (VP2 and VP3). All other testing procedures and subject controls matched the primary study.

### 2.3. Equipment/Instruments

All exercise tests were conducted on a Monark 839E electronically braked cycle ergometer (Varberg, Sweden), which can continuously adjust brake force in response to changes in pedaling cadence to keep power output constant. Seat height and handlebar position were recorded for each subject and was replicated for subsequent visits. Pulmonary ventilation and gas exchange were measured during all visits with a Parvomedics metabolic cart (Truemax2400, Sandy, UT, USA). The mean of the highest 2 consecutive 15 s averages was used to determine VO_2max_ in all tests. Heart rate max was determined using the highest 15 s average at the end of the test. Standard 3-point calibration was performed before each test or every 4 h per manufacturer recommendation. Heart rate was continuously measured using a Polar heart rate monitor (Lake Success, NY, USA) and all blood lactate measurements were done with a Lactate Plus blood lactate analyzer (Sports Resource Group, Minneapolis, MN, USA).

### 2.4. Exercise Testing

The following procedures were used during all cycle ergometer tests. Ratings of perceived exertion were taken at the end of each minute. Capillary blood lactate was measured via finger-prick at rest prior to testing and then again immediately following exhaustion. Subjects were given verbal encouragement throughout the test and when pedal rate slowed below 50 revolutions per min (rpm) for more than 5 s, the test was stopped and time to exhaustion (TTE) was recorded.

The GXT protocol was chosen using an estimated VO_2max_, estimated max wattage, and a grade designed to reach max wattage in approximately 10 min following the warm-up [17]. During the warm up, subjects were instructed to choose a 10 rpm range of their comfort between 60 and 90 rpm. Once the ramp-style GXT test began, the wattage increased from the warm up wattage continuously throughout the test. Max wattage was selected as the final set power output when pedal rate slowed below 50 rpm.

The max wattage on the GXT was used to determine the VP wattage based on the assigned percentage. In the primary study, the order of the 4 VP visits was randomly selected to avoid a sequence effect and the set wattage was hidden from the subjects. Subjects were notified of the oncoming increase in resistance of the cycle during the last 5 s of the warm-up and wattage was immediately increased to the set chosen VP wattage. Subjects were instructed to pedal as long as possible at an rpm range (60–70, 70–80, or 80–90 rpm) of their choosing. This rpm range was recorded after the first VP test and repeated for all subsequent tests.

### 2.5. Statistical Analyses

*p* values were two-tailed, and values of ≤0.05 were considered statistically significant. All data are presented as mean ± SD unless otherwise noted. Descriptive statistics for the subjects who completed the primary study were calculated by sex (Table 1). Linear mixed effects models (LME) were used in lieu of repeated measures due to their superior ability to deal with unequal groups and their lack of reliance on sphericity, equal variance and covariance assumptions [28,29]. An LME model was fitted to each of the outcome measures to assess the effects of test, sex, and the interaction test X sex, with a random effect to control for subject using the maximum likelihood method. Post-hoc tests were run using Bonferroni correction, comparing the means between each pair of verification tests and the GXT. Pearson’s correlations were used to determine the relationships between time on each test and differences in VO_2max_ and HR_max_, as well as the difference between the highest VO_2max_ achieved and each VP VO_2max_ with the TTE on each VP test. All tests were implemented in R [25]. Mixed effects models were fitted using the lme4 package [30] with lmerTest [31] to compute the *p*-values, while post-hoc tests were conducted using the emmean package [32].

For the reliability study, we used intraclass correlation coefficients (ICC, one way, consistency) and coefficient of variation (CV) to assess and examine reliability. A Bland-Altman Plot was used to determine the relationship between mean VO_2max_ during VP 2 and VP 3 and the highest VO_2max_ achieved. Paired samples t-tests were used to assess differences between VP 2 and VP 3 VO_2max_, TTE, HR_max_, RER_max_, wattage, and the difference between GXT VO_2max_ and VP VO_2max_.

## 3. Results

### 3.1. Primary Study

Of the 34 enrolled subjects (15 males and 19 females), 4 females dropped out due to time constraints or inability to comply with the 48 h of rest requirement between each test, resulting in 15 males and 16 females (See Table 1) completing all requirements of the study. All outcome variables during the GXT and VP tests are shown in Table 2 (males) and Table 3 (females).

See Figure 1 and Table 2 and Table 3 for the VO_2max_ data for both males and females. Results of the mixed effect model comparing all 4 VPs and the GXT showed a significant difference in VO_2max_ between males and females (Males: 3.61 ± 0.60, Females: 2.33 ± 0.32 L/min; *F* (1,31) = 60.3, *p* < 0.001) and a significant test x sex interaction (*F* (4,124) = 3.10, *p* = 0.02, See Table 2 and Table 3, and Figure 1). There was no difference between the 5 mean VO_2max_ values with males and females combined (*F* (4,124) = 1.97, *p* = 0.10). When males and females were separated there was a significant main effect for test (*F* (4,60) = 2.81, *p* = 0.03) with the males, but not with the females (*F* (4,64) = 1.49, *p* = 0.22); 12 of the 15 males achieved their highest VO_2max_ during a submaximal VP, but the females highest achieved VO_2max_ values were more evenly distributed across the 4 VP work rates. Post-hoc tests for the males revealed the GXT elicited a significantly higher VO_2max_ compared to the VP at 105% (GXT −105%: mean difference = 0.14 ± 0.05, *t* (133) = 2.83, *p* = 0.05) but was not different from the other VPs. Additionally, there were significant differences between the 80% and 90% when compared to 105% (105–80%: mean difference = −0.17 ± 0.05, *t* (133) = 3.51, *p* < 0.01; 105–90%: mean difference = −0.16 ± 0.05, *t* (133) = 3.21, *p* = 0.02).

Pearson correlations revealed no relationship between TTE on the VP tests and VO_2max_ (r = 0.01, *p* = 0.94). There were significant relationships with the difference between the highest VO_2max_ and VP VO_2max_ correlated with TTE on each VP test (females: r = 0.39, *p* = 0.002; males: r = −0.32, *p* = 0.01).

### 3.2. Reliability Study

Of the 23 males and females enrolled in the reliability study, 20 (10 females, 10 males; BMI = 22.7 ± 2.9 kg/m^2^; Age = 25.5 ± 4.0 year) completed all the requirements for their data to be analyzed. The Bland-Altman plot comparing the highest VO_2max_ achieved and mean difference between VP 3 and VP 2 showed no bias and small limits of agreement (mean difference of 0.03 L/min, 95% limits of agreement: 0.17 and −0.11 L/min, see Figure 2). ICCs and CVs for males and females combined showed excellent consistency for VP VO_2max_ (ICC = 0.991; CV = 2.68 ± 2.52%). ICC and CV for female participants VP VO_2max_ demonstrated excellent consistency (ICC = 0.987; CV = 2.5%). Male participants VP VO_2max_ displayed excellent consistency (ICC = 0.941; CV = 2.2%). ICCs and CVs for males and females combined showed excellent consistency for GXT VO_2max_ (ICC = 0.989; CV = 2.7%). Wattage and TTE did significantly differ between VP 2 and VP 3, but VO_2max_, HR_max_, RER_max_, and the difference between GXT and VP VO_2max_ did not (See Table 4).

## 4. Discussion

The primary finding of this study is that the supramaximal verification phase test (105%) elicited a significantly lower VO_2max_ value compared to the submaximal VP tests (80 and 90%) and the GXT in males only. Conversely, there were no significant differences across the VP intensities and GXT for our female subjects and 80% of the males achieved their highest VP VO_2max_ during a submaximal VP test compared to only 37.5% of the females. Our study is the first to compare four different VP intensities and to evaluate differences between males and females. Furthermore, our reliability study showed excellent consistency, no bias, and acceptable limits of agreement across a range of VO_2max_ values for the 90% VP test in both males and females.

The results of the present study partially agree with those of Astorino [20]. Study number two in the Astorino [20] paper found a significant test x gender interaction where the males had a higher VO_2max_ on the incremental test compared to the supramaximal VP and the females did not. Conversely, study number one showed no test x gender effect but overall showed a significantly higher VO_2max_ in the incremental compared to the VP. Similar to study two above, we found a significant test x sex interaction showing no difference across VP intensities for females but a significantly lower VO_2max_ in the 105% VP compared to GXT, 80%, and 90% VP. Our results also agree with a previous study that used 4 square wave bouts of exercise to exhaustion in males and females in order to determine critical power [33]. These results show that the supramaximal square wave bout (118 and 122% of GXT power in males and females, respectively) only elicited 91% of VO_2max_ achieved on the GXT in males, but 105% of GXT VO_2max_ in females. Similarly, the highest mean VO_2peak_ values for the males (97% of GXT VO_2max_) were achieved during two submaximal square-wave bouts (P2 and P3, 80 and 71% of GXT peak power output, respectively). For the females, the highest VO_2peak_ values during the square-wave bouts were achieved at power outputs equal to 90 and 122% of GXT peak power eliciting 108% and 105% of GXT VO_2max_, respectively. This comparison should be viewed with caution due to the lower subject number (9 males and 4 females), a lack of statistically significant differences between VO_2peak_ values, and differences in power outputs relative to GXT peak power.

The reason for a difference in males and females in optimal VP intensity is not apparent. One possible reason may be bias induced via calculation of VP work rate based on a percentage of GXT maximum work rate. Our mean peak wattage was 83 watts higher in males compared to females. The 5% above GXT added on average 15 watts for the males, but only 10 watts for the females and this extra added wattage could be enough to push male participants above the severe domain of exercise into the extreme domain where TTE is too short to elicit VO_2max_ [19,34]. Along these same lines another explanation would be faster or similar VO_2_ kinetics in females compared to males coupled with similar times to exhaustion. As expected in our study, the mean male VO_2max_ was ~10 mL/kg/min higher than the female even though the TTE was similar. Accordingly, the males in our study may have not had sufficient time to reach their higher VO_2max_ during the 105% VP, whereas the females did. An extreme example of this in our study was found in our subject with the highest VO_2max_ (5.9 L/min) who had similar TTE on the 100 and 105% VP tests when compared to the other males, but those VP tests produced much lower VO_2max_ values compared to his lower intensity VP tests. That subject’s data can be seen as the 2 outliers in panel A in Figure 3 below. This is potentially supported by previous work showing no gender effect on gross efficiency or delta efficiency during submaximal cycling [35]. Importantly, the efficiency measured in that study [35] was submaximal and may not hold true at maximal and supramaximal work rates. Hill et al. [19] have suggested that times to exhaustion of at least 2 min may be necessary to elicit VO_2max_. The average TTE during the 105% VP in our males was 2.44 min but did not elicit VO_2max_. The study discussed above by Sawyer et al. [33] showed that VO_2max_ reached in as little as 1.27 min in females, but 1.25 min only elicited 91% of GXT VO_2max_ in males. Similarly, the current study shows a mean TTE during the 105% VP in females of 2.19 min, which produced a VO_2max_ not different to GXT VO_2max_. Interestingly (as shown in Figure 3), the correlations between the difference in the highest VO_2max_ and each VP VO_2max_ with TTE show a positive relationship for females (r = 0.39) and a negative one for males (r = −0.32). To clarify, for males the shorter the VP duration the further their elicited VO_2_ was from VO_2max_, but for females the shorter the VP duration the closer their elicited VO_2_ was to VO_2max_. These correlations are reflective of the maximal and supramaximal VPs producing the highest VO_2max_ in the females and the submaximal VPs producing the highest VO_2max_ in males. This sex difference could also be explained by the fact that our female subjects may have been more fit than the males and could withstand severe intensity exercise for longer. This hypothesis is supported by the fact that for some of our female subjects the 80% VP was at or below critical power due to exhaustion never occurring in the 20 min test, but all males reached exhaustion in under 10 min. Furthermore, the average TTE during the 80% VP was 9.24 min in the females and only 6.28 min in the males. Comparison of our VO_2max_ data to normative values does not support this hypothesis since the mean VO_2max_ for females and males were both in the high end of the “fair” category [36]. Critical power would be a better indicator of fitness in terms of maximally sustainable VO_2_ [34], but we did not measure critical power in this study.

In addition, our finding that a submaximal VP test produces higher VO_2max_ values in males may be limited to cycle ergometer testing. It has been shown that cycle ergometer VO_2max_ is typically 5–20% lower than treadmill VO_2max_ due to regional muscle fatigue [37]. The regional muscle fatigue is most likely why the supramaximal VP leads to shorter times to exhaustion. To our knowledge no study has shown supramaximal VP testing on a treadmill to elicit lower VO_2max_ than submaximal or GXT. The larger muscle mass involved in locomotor exercise may allow for supramaximal VP testing on the treadmill to produce adequate times to exhaustion and therefore higher VO_2max_ values compared to cycle ergometer.

The results of the reliability study further establishes support for using a 90% VP test by showing excellent reliability (ICC = 0.991), extremely similar VO_2max_ values with repeat testing (VP2 VO_2max_: 3.01 ± 0.69 L/min, VP 3 VO_2max_: 3.04 ± 0.69 L/min; *p* = 0.55), a small coefficient of variation (overall CV = 2.68%), no bias across a wide range of VO_2max_ values (mean difference between VP2 and VP2: 0.03 L/min), and acceptable limits of agreement. Previous work [5] has shown day to day variation in the measurement of VO_2max_ to be ~5% (0.23 L/min with mean VO_2max_ of 4.51 L/min), therefore our limits of agreement of ~3–6% of mean VO_2max_ (See Figure 2; 0.17 and −0.11 L/min; ~4% of VO_2max_) show the random error between repeat tests to be in an acceptable range. Our ICC and CV values are similar to previous studies using supramaximal VP tests [15,17,38,39,40,41]. Furthermore, these results were replicated when we examined the males and females separately (See Table 4). To our knowledge, the current study is the first to examine the reliability of a 90% power output submaximal verification phase to confirm VO_2max_.

### Strengths and Limitations

One strength of this study is the inclusion of both males and females with adequate statistical power to draw conclusions for each sex independently. Furthermore, we are not aware of any other studies that have specifically compared as many different VP intensities as compared in this study. Moreover, using LME models to compare repeated measures within subjects afforded us the ability to compare multiple repeated tests without weakening the analyses due to inequality of variance, covariance, or sphericity. One possible limitation of this study is the fact that the VP tests were completed on separate days from the GXT. Most common use of VP testing would include GXT followed by a VP in one session, but completing the VP on a separate day has shown similar results [9,38]. The RER_max_ values for males in the primary study were higher than expected for maximal exercise, but not out of the range of values seen in previous work, especially with short times to exhaustion [9,42,43]. Importantly, use of a submaximal intensity for VP testing does not extend the VO_2_-power relationship to construct a VO_2_ plateau as is recommended by Poole and Jones [11] with supramaximal VP testing. We did not attempt to extend the VO_2_-power relationship but to find a VP intensity that produces the highest VO_2max_. In regards to the reliability study our results can only be applied to the VP test at 90% GXT wattage and therefore do not have implications about the reliability or bias in the other VP intensities. Lastly, this study has implications only for cycle ergometer testing in young healthy adults, whereas different conclusions may be reached with other modes of exercise in different populations.

## 5. Conclusions

In conclusion we found that males and females had different VO_2_ responses to cycle ergometry verification phase testing. In males the submaximal tests elicited higher VO_2max_ values compared to supramaximal testing. For females, a range of intensities (80–105%) produce similar VO_2max_ values but the unnecessarily high times to exhaustion in the 80% VP do not warrant its use. We also established excellent reliability, low repeated test variability, and lack of bias across a range of VO_2max_ values using the 90% VP in both males and females. Our results show that submaximal VP testing should be considered an acceptable method for verification of VO_2max_ and potentially the preferable method in young healthy males during cycle ergometry.

## Figures and Tables

**Figure 1 sports-08-00163-f001:**
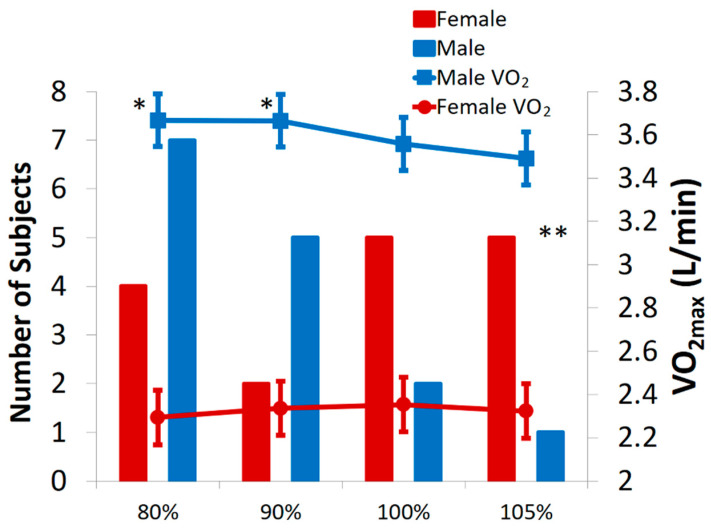
Primary axis = Number of subjects that elicited their highest VO_2max_ during each of the four verification phase tests. Secondary axis = Mean ± SD VO_2max_. * Significantly higher than VP 105%. ** Significant interaction VP intensity x Sex (*p* = 0.02).

**Figure 2 sports-08-00163-f002:**
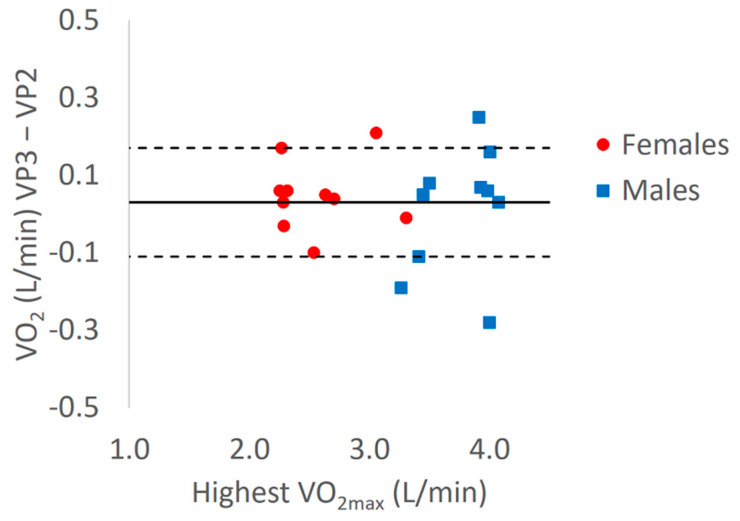
VP VO_2max_ Bland-Altman plot. Y axis = VO_2_ (L/min) Verification Phase 3 − Verification Phase 2; X axis = Highest VO_2max_ (L/min); Dashed lines = Mean ± 1.96 * SD; Solid line = mean of VP3 − VP2 (0.03 L/min).

**Figure 3 sports-08-00163-f003:**
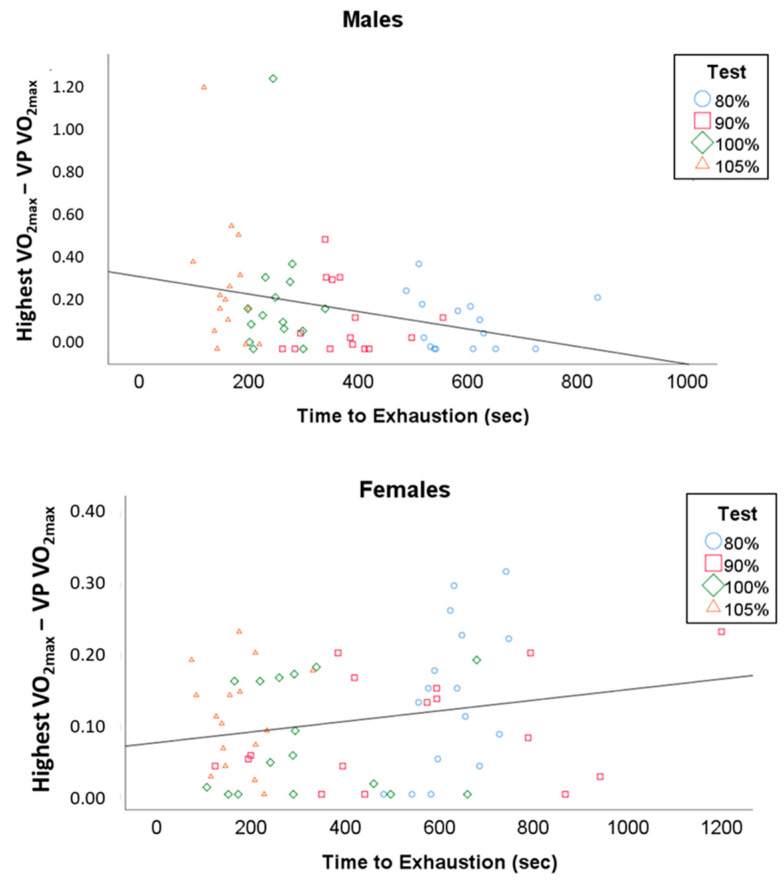
Scatterplots showing the relationship between time to exhaustion and the difference between the highest VO_2max_ and each verification phase (VP) VO_2max_. Each VP test is shown by a different color and marker type. Top panel = males (r = −0.32, *p* = 0.01); Bottom panel = females (r = 0.39, *p* = 0.002).

**Table 1 sports-08-00163-t001:** Descriptive and anthropometric data for all subjects who completed the primary study.

	Males (n = 15)	Females (n = 16)
Age (years)	22 ± 2	21 ± 2
Height (cm)	178.1 ± 8.4	172.0 ± 6.4
Weight (kg)	77.3 ± 7.5	63.1 ± 10.7
BMI (kg/m^2^)	24.5 ± 2.2	23.2 ± 3.3
VO_2max_ (mL/kg/min)	48.82 ± 6.32	39.12 ± 5.96

Values represent mean ± SD. VO_2max_ represents highest of all 5 tests for each subject.

**Table 2 sports-08-00163-t002:** Outcome variables for the males for each test.

	VO_2max_ (L/min)	HR_max_ (bpm)	RER_max_	Blood Lactate (mmol/L)	Peak Power (W)	Total Time (min)	RPE
**GXT**	3.64 * ± 0.67	184 ± 12	1.27 ± 0.07	12.9 ± 2.3	297 ± 60	9.90 ± 1.62	19 ± 1
**80%VP**	3.67 * ± 0.71	183 ± 13	1.25 ± 0.08	13.1 ± 2.4	238 ± 48	6.28 ± 1.28	20 ± 1
**90%VP**	3.66 * ± 0.67	181 ± 12	1.33 ± 0.10	12.8 ± 1.8	268 ± 54	4.21 ± 0.69	20 ± 1
**100%VP**	3.56 ± 0.51	180 ± 12	1.41 ± 0.08	13.8 ± 1.9	297 ± 60	2.70 ± 0.53	20 ± 1
**105%VP**	3.50 ± 0.47	177 ± 12	1.48 ± 0.12	12.9 ± 1.1	312 ± 63	2.44 ± 0.40	20 ± 0

GXT = Initial graded exercise test; VP = verification phase test. Values represent mean ± SD; * Significantly higher than 105% VP.

**Table 3 sports-08-00163-t003:** Outcome variables for the females for each test.

	VO_2max_ (L/min)	HR_max_ (bpm)	RER_max_	Blood Lactate (mmol/L)	Peak Power (W)	Total Time (min)	RPE
**GXT**	2.35 ± 0.33	186 ± 6	1.22 ± 0.06	11.2 ± 2.0	214 ± 26	10.44 ± 1.24	19 ± 1
**80%VP**	2.29 ± 0.34	187 ± 7	1.18 ± 0.12	11.2 ± 2.2	171 ± 21	9.24 ± 5.00	19 ± 1
**90%VP**	2.34 ± 0.33	185 ± 5	1.23 ± 0.09	12.4 ± 2.1	192 ± 23	5.33 ± 2.85	19 ± 1
**100%VP**	2.36 ± 0.32	185 ± 4	1.32 ± 0.08	11.8 ± 1.7	214 ± 26	2.86 ± 1.07	18 ± 2
**105%VP**	2.32 ± 0.32	184 ± 7	1.33 ± 0.09	12.0± 1.6	224 ± 27	2.19 ± 0.59	19 ± 1

GXT = Initial graded exercise test; VP = verification phase test. Values represent mean ± SD.

**Table 4 sports-08-00163-t004:** Outcome variables for all participants for each verification phase (VP) test for the submaximal VP reliability study.

	VP 2			VP 3			VP2 vs. VP3 All Subjects
	All	Males	Females	All	Males	Females	Significance
**VO_2 mL_/kg/min**	40.91 ± 5.93	43.29 ± 5.48	38.52 ± 5.62	41.37 ± 6.24	43.55 ± 6.79	39.2 ± 5.21	0.22
**VO_2_ L/min**	3.01 ± 0.69	3.59 ± 0.31	2.43 ± 0.39	3.04 ± 0.69	3.54 ± 0.37	2.48 ± 0.39	0.31
**HR_max_**	168 ± 41	176 ± 11	161 ± 57	179 ± 10	178 ± 10	180 ± 10	0.27
**RER_max_**	1.13 ± 0.09	1.13 ± 0.09	1.13 ± 0.09	1.14 ± 0.09	1.15 ± 0.10	1.13 ± 0.08	0.69
**TTE**	2.68 ± 0.90	2.24 ± 0.68	3.13 ± 0.90	2.29 ± 0.50	2.05 ± 0.57	2.52 ± 0.28	0.02 *
**Wattage**	235 ± 54	275 ± 40	195 ± 32	240 ± 55	280 ± 43	200 ± 33	0.04 *
**VO_2_ (L/min): VP-GXT**	−0.02 ± 0.13	−0.05 ± 0.16	0.00 ± 0.09	−0.01 ± 0.14	−0.03 ± 0.17	0.02 ± 0.09	0.55

* *p* < 0.05.

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
