# Peer review of "Supra-Versus Submaximal Cycle Ergometer Verification of VO2max in Males and Females"

_sports, 2020, doi:10.3390/sports8120163_

Round 1

Reviewer 2 Report

The present study is relevant, interesting, with possibilities for application in the field. Contributes with new knowledge in this scientific area;

The sample is not very large but it seems sufficient to be able to draw some important conclusions; As the authors refer, these results and conclusions are specific to the cycle ergometer and to groups of medium performance level subjects; Comparisons between genders must be viewed with reserve, as the average level of men may be different from that of women; What will happen to elite athletes? Although not the objective of this study, it would be important to know to be able to help coaches and athletes. Maybe in another study ?!
Another interesting aspect to study better may be the definition of the intensity levels used in the comparison between the tests; If, eventually, the average level of the groups is different, it might be justified to use critical power as a criterion; Perhaps defining levels between critical power and maximum aerobic power. !
In general, the study's conclusions may have specific application in the context of prescription and training in a cycle ergometer or cycling.

Author Response

Dear Reviewer,

Thank you for your comments and suggestions for future research. We too agree that the addition of critical power to this study would have been ideal and could add a lot to the interpretation. We plan to implement this in future work. Furthermore, we plan to implement similar studies in different populations to extend its implications. 

Thank you for your time in reviewing our paper. 

Brandon Sawyer, PhD

Round 2

Reviewer 1 Report

The authors have satisfactorily addressed all my comments.

This comment is based on the newly included Figure 3 (Scatter plots for relationship between TTE and Difference in highest VO2 and verification test VO2). There are two datapoints in the males scatter plot that are clearly outliers because they are two times the maximum value for males and 3 times and maximum value for females. I would request that the authors:

1) Check that the quality of data for the 100 and 105% tests for this subject (or for the two subjects if these are not from the same subject) were good (the breath by breath data are not showing drops to zero that may indicate a poor mask seal). It seems like a good test because TTE is within the range with all the other subjects; just the VO2 difference is quite large.

2) Confirm that their results for males will not change if they remove these outlier data points or justify why these may be random variation and do not warrant removing/reanalysis. At least from the plot it appears that removing these outliers could weaken the TTE vs. difference in VO2 correlation but I don’t know if removing these will change the final results (80 and 90% VO2 > 105%).  
